

# Multidecadal ozone trends in China and implications for human
# health and crop yields: A hybrid approach combining chemical
# transport model and machine learning
Jia Mao[1], Amos P. K. Tai[1,2,3], David H. Y. Yung[1], Tiangang Yuan[1], Kong T. Chau[1], and Zhaozhong
Feng[2,4]
[1] Earth and Environmental Sciences Programme and Graduate Division of Earth and Atmospheric Sciences, Faculty of
Science, The Chinese University of Hong Kong, Sha Tin, Hong Kong SAR, China
[2] Collaborative Innovation Center on Forecast and Evaluation of Meteorological Disasters (CIC-FEMD), Nanjing
University of Information Science and Technology, Nanjing 210044, Jiangsu, China
[3] State Key Laboratory of Agrobiotechnology, and Institute of Environment, Energy and Sustainability, The Chinese
University of Hong Kong, Hong Kong SAR, China
[4] Key Laboratory of Ecosystem Carbon Source and Sink, China Meteorological Administration (ECSS-CMA), Nanjing
University of Information Science & Technology, Nanjing, 210044, China
*Correspondence to*: Amos P. K. Tai (amostai@cuhk.edu.hk)
**Abstract.** Surface ozone ($O_3$) is well known to pose significant threats to both human health and crop production worldwide.
However, a multi-decadal assessment of $O_3$ impacts on public health and crop yields in China is lacking due to insufficient
long-term continuous $O_3$ observations. In this study, we used a machine learning (ML) algorithm to correct the biases of
$O_3$ concentrations simulated by the chemical transport model from 1981–2019 by integrating multi-source datasets. The
ML-enabled bias correction offers improved performance in reproducing observed $O_3$ concentrations, and thus further
improves our estimates of $O_3$ impacts on human health and crop yields. Our results show that a warm-season increasing
trend of $O_3$ in Beijing-Tianjin-Hebei and its surroundings (BTHs), Yangtze River Delta (YRD), Sichuan Basin (SCB) and
Pearl River Delta (PRD) regions are 0 .32 μg m$^{-3}$ yr$^{-1}$, 0.63 μg m$^{-3}$ yr$^{-1}$, 0.84 μg m$^{-3}$ yr$^{-1}$, and 0.81 μg m$^{-3}$ yr$^{-1}$ from 1981
to 2019, respectively. In more recent years, $O_3$ concentrations experience more fluctuations in the four major regions. Our
results show that only BTHs have a perceptible increasing trend of 0.81 μg m$^{-3}$ yr$^{-1}$ during 2013–2019. Meteorological
factors play important roles in modulating the interannual variability of surface $O_3$, wherein synoptic systems (e.g., high-
pressure system, Western Pacific subtropical high, tropical cyclone) are closely related to the spatiotemporal distribution
of regional $O_3$ via influencing regional weather conditions and transport processes. Using AOT40-China dose-yield
relationship, the estimated relative yield losses (RYLs) for wheat, rice, soybean and maize are 17.6%, 13.8%, 11.3% and
7.3% in 1981, and increases to 24.2%, 17.5%, 16.3% and 9.8% in 2019, with an increasing rate of +0.03% yr$^{-1}$, +0.04%
yr$^{-1}$, +0.27% yr$^{-1}$ and +0.13% yr$^{-1}$, respectively. The estimated annual all-cause premature deaths induced by $O_3$ increase
from ~55,900 in 1981 to ~162,000 in 2019 with an increasing trend of ~2,980 deaths yr$^{-1}$. The annual premature deaths
related to respiratory and cardiovascular disease are ~34,200 and ~40,300 in 1998, and ~26,500 and ~79,000 in 2019,
having a rate of change of –546 and +1,770 deaths yr$^{-1}$ during 1998–2021, respectively. Our study, for the first time, used
ML to provide a robust dataset of $O_3$ concentrations over the past four decades in China, enabling a long-term evaluation
of $O_3$-induced crop losses and health impacts. These findings are expected to fill the gap of the long-term $O_3$ trend and
impact assessment in China.





## 1 Introduction

Surface ozone ($O_3$), an important secondary air pollutant, is mainly generated through photochemical reaction of volatile organic compounds (VOCs), carbon monoxide (CO), and nitrogen oxides ($NO_x$) in the presence of sunlight. As a strong oxidant, $O_3$ at the ground level is detrimental to human health and vegetation. More recently, due to the rapid urbanization and industrialization, the summertime $O_3$ pollution has become an emerging concern in China. Li et al. (2020) reported that the mean summer 2013–2019 trend in maximum daily 8-h average surface $O_3$ (MDA8-$O_3$) was +1.9 ppb yr$^{-1}$ in China, with high values widely observed in the North China Plain (NCP), Yangtze River Delta (YRD), and Pearl River Delta (PRD) regions. On the regional scale, the exposure of humans and vegetation to $O_3$ is greater in China than in other developed regions of the world (Lu et al., 2018). Several studies have suggested the important roles of climate and land cover changes on $O_3$ pollution in addition to anthropogenic emissions (Fu and Tai, 2015; Wang et al., 2020). It has been suggested that global warming and the changing land use may further increase surface $O_3$ by the late 21$^{st}$ century (Kawase et al., 2011; Wang et al., 2020), which can pose greater threats to human health and food security.

Meteorological factors can modulate the temporal and spatial patterns of $O_3$ via affecting the physical and chemical processes within the atmosphere (Liu et al., 2019; Mao et al., 2020; Yin and Ma, 2020). High temperature, low relative humidity and low planetary boundary height are conducive to the photochemical production and $O_3$ accumulation. Jacob and Winner (2009) summarized that the enhanced $O_3$ levels at higher temperatures are primarily driven by increased biogenic VOC emissions from vegetation and reduced lifetimes of peroxyacetyl nitrate (PAN) due to accelerated decomposition of PAN into $NO_x$. Besides, the changes in wind speed and direction can affect $O_3$ concentrations through transport. Land cover and land use change affects $O_3$ air quality by perturbing surface fluxes, hydrometeorology, and concentrations of atmospheric chemical components (Tai et al., 2013; Fu and Tai, 2015; Liu et al., 2020; Ma et al., 2021). For instance, the terrestrial biosphere is a major source of isoprene, which plays a significant role in modulating $O_3$ concentrations. In the Intergovernmental Panel on Climate Change (IPCC) A1B scenario, Tai et al. (2013) found that widespread crop expansion could reduce isoprene emission by ∼10 % globally compared with the present land use. Such a reduction could decrease $O_3$ by up to 4 ppb in the eastern US and increase $O_3$ by up to 6 ppb in South and Southeast Asia, whereby the difference in the sign of responses is primarily determined by the different $O_3$ production regimes.

The increasing health burden due to air pollution has become an important contributor to global disease burden. Some recent studies have demonstrated that short-term $O_3$ exposure negatively impacts human health, especially via respiratory, and cardiovascular mortality (Shang et al., 2013; Yin et al., 2017b; Feng et al., 2019; Zhang et al., 2022a). In 2015–2018, the estimated annual total premature mortality related to $O_3$ pollution in 334 Chinese cities was 0.27 million for 2015, 0.28 million for 2016, 0.39 million for 2017, and 0.32 million for 2018 (Zhang et al., 2021). Maji and Namdeo (2021) reported that short-term all-cause, cardiovascular and respiratory premature mortalities attributed to the ambient 4$^{th}$ highest MDA8-$O_3$ exposure were 156,000, 73,500 and 28,600 in 2019, showing increases of 19.6%, 19.8% and 21.2%, respectively, compared to 2015. Zhang et al. (2022b) reported that each 10 μg m$^{-3}$ increase in the MDA8-$O_3$ can lead to a rise of 0.41 % (95 % CI: 0.35 %–0.48 %) in all-cause, 0.60 % (95 % CI: 0.51 %– 0.68 %) in cardiovascular and 0.45 % (95 % CI: 0.28 %– 0.62 %) in respiratory mortality.

The damage to plants induced by $O_3$ is mainly caused by the stomatal uptake of $O_3$ into the leaf interior instead of direct plant surface deposition (e.g., Clifton et al., 2020). In previous studies, a variety of concentration-based metrics have been widely used to assess the $O_3$ risks to crop yield and ecosystem functions. Initially, a 7-hour (09:00–15:59) mean metric (M7) was proposed, which was later extended to a 12-hour (08:00–19:59; referred to M12) to include late-day $O_3$ concentrations. Cumulative metrics have also been developed to evaluate the impacts of $O_3$ on crops. The accumulated $O_3$ over a threshold of 40 ppb (AOT40) is a widely used metric to evaluate the phytotoxic effects of $O_3$. Compared to AOT40 using a linear function, another metrics, W126, considers the nonlinear response of yield loss to $O_3$ exposure whereby higher $O_3$ concentrations will progressively induce more severe yield losses. However, many studies have suggested that the stomatal uptake of $O_3$ is more related to vegetation damage than $O_3$ exposure per se (Feng et al., 2012; Feng et al., 2018;



Pleijel et al., 2022). In the recent two decades, the flux-based approach therefore has been developed and increasingly used
to assess the relationships between the stomatal $O_3$ uptake and crop yields. Tai et al. (2021) compared the results of the
estimated global crop yield losses using three concentration-based and two flux-based $O_3$ exposure metrics, and showed
that the concentration-based metrics differ greatly among themselves, while the two flux-based metrics are generally close
to each other, which lie close to the middle of the range covered by all metrics.

At present, a comprehensive long-term assessment of $O_3$ impacts is hindered by a lack of continuous $O_3$ observations
in China (Lu et al., 2018; Gong et al., 2021). From both health and food perspectives, reliable long-term estimates of $O_3$
are critically needed to better understand the $O_3$ damage over the past few decades since the beginning of rapid industrial
transformation in the 1980s. In previous studies, various alternative approaches have been used to address the problem of
insufficient observations. The multiple linear regression (MLR) model is often used for extrapolation to construct
spatiotemporal distributions of air pollutants (Moustris et al., 2012; Abdullah et al., 2017). However, the linear statistical
methods are generally limited by their incapability to capture the nonlinear relationships between air pollutants and
precursors as well as meteorological fields. Chemical transport models (CTMs), based on mathematical representation of
atmospheric physical and chemical processes, are also the common tool to simulate air pollutant concentrations
spatiotemporally (Fusco and Logan, 2003; Liu and Wang, 2020b; Wang et al., 2022a). Taking the advantages of the CTM,
Fu and Tai (2015) investigated the impacts of historical climate and land cover changes on tropospheric $O_3$ in East Asia
between 1980 and 2010. However, the utility of CTMs is often limited by their high computational cost when conducting
long-term simulations at high spatiotemporal resolutions. Large biases also exist due to uncertainties in historical emission
inventories, parameterization of physical and chemical processes, and initial and/or boundary conditions, and these errors
tend to increase at finer spatiotemporal scales.

In recent years, machine learning (ML) methods have gained increasing popularity in air pollution studies (Liu et al.,
2020; Ma et al., 2021). In the early stage of applying ML to atmospheric chemistry, ML methods were usually used as an
independent method from CTMs (Hu et al., 2017; Zhan et al., 2017), for instance, to predict $O_3$ concentrations by mapping
the nonlinear relationships between observed $O_3$ concentrations and their possible shaping factors. These applications are
usually purely data-driven, whereby the ML algorithms do not involve any representation of the physical mechanisms
behind the relevant processes. With powerful algorithms and user-friendly hyperparameter tuning processes, some well-
trained ML models, driven by data from multiple sources including reanalysis and satellite data, have shown even higher
predictive capacity than process-based models.  The advantages of ML methods over CTMs include more flexible choices
for input data and spatiotemporal resolution, and substantially lower computational costs (Bi et al., 2022). However, purely
data-driven ML methods are known to suffer a lack of transparency and interpretability, which renders it more difficult to
offer adequate scientific interpretation for the physical mechanisms behind. Thus, a hybrid approach combining ML
algorithms and CTM-simulated results have been increasingly used to predict air pollutants and understand their trends in
recent years. Integrating data from various sources, ML methods have been used as a tool to correct the biases in the lower-
resolution simulated results from CTMs (Di et al., 2017; Ivatt and Evans, 2020; Ma et al., 2021). Based on process-based
CTMs integrating decades of accumulated knowledge in Earth system science, while taking advantage of ML to address
still-existing model errors, the hybrid approach has great potential in tackling air quality problems (Irrgang et al., 2021).

In this work, we incorporated the $O_3$ concentrations directly simulated by the Goddard Earth Observing System
coupled with Chemistry (GEOS-Chem) model at a lower resolution into a bias-corrected, finer-resolution dataset by
integrating them with $O_3$ observations from 2016 to 2018 (for validation purpose), high-resolution metetrological fields,
land use data and other geographical information from multiple sources using a tree-based ML algorithm, LightGBM. The
final high-resolution hourly $O_3$ dataset with a resolution of 0.25°×0.25° from 1981 to 2019 was further used to assess the
impacts of $O_3$ on human health and crop yields over the past four decades. The simultaneous analysis of the combined
impacts of $O_3$ on agriculture and human health can offer more comprehensive policy implications for the mitigation of $O_3$-
related impacts across China.



**2 Data and methods**

**2.1 Air quality, meteorological, land and crop data**

Hourly surface $O_3$ observations (µg m$^{-3}$) from 2016 to 2018 were obtained from the China National Environment Monitoring Center Network (http://106.37.208.233:20035/) established by the Ministry of Ecology and Environment of China. The MDA8-$O_3$ of each site was calculated with at least 14 valid hourly values from 08:00 to 24:00 local time. A total of 1016 sites were selected after deleting the missing and abnormal data (**Fig. 1**).

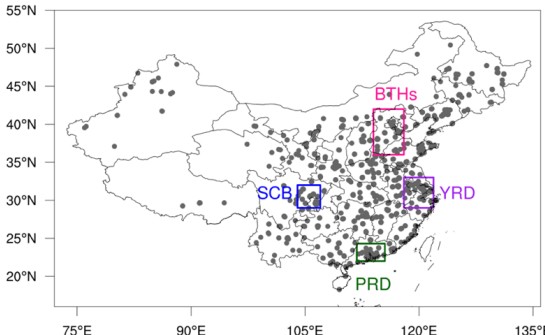

**Figure 1. Study domain and locations of the selected monitoring sites. The pink, blue, purple and green rectangles indicate the Beijing-Tianjin-Hebei and its surroundings (BTHs), Sichuan Basin (SCB), Yangtze River Delta (YRD), and Pearl River Delta (PRD) regions, respectively, for more detailed analysis.**

The surface meteorological fields used in this study include sea surface pressure, horizontal wind at 10 m, air temperature at 2 m, downward solar radiation, surface albedo, and total precipitation. The variables selected at 850 hPa and 100 hPa include relative humidity, horizontal and vertical velocity. These meteorological variables have been shown by many previous studies to correlate strongly with surface $O_3$ concentrations as discussed above. Hourly reanalysis data for meteorological variables were obtained from the fifth generation European Center for Medium-Range Weather Forecasts (ECMWF) reanalysis dataset (ERA5) with a spatial resolution of 0.25°×0.25° from 1981 to 2019 (https://cds.climate.copernicus.eu/). This spatial resolution sets the highest limit of resolution for our hybrid $O_3$ product.

The national land use data with a spatial resolution of 1 km×1 km for 2013 were obtained from the Resource and Environment Science Data Center of the Chinese Academy of Sciences (RESDC) (http://www.resdc.cn). Six primary types of land use are considered: cultivated land, forestland, grassland, water bodies, construction land, and unused land. Nationwide elevation data were also provided by the RESDC (https://www.resdc.cn/data.aspx?DATAID=123), which is resampled based on the latest Shuttle Radar Topography Mission (SRTM) V4.1 data developed in 2000.

The spatial distribution of the harvested areas for four staple crops (wheat, rice, maize, soybean) for China was obtained from the Global Agro-Ecological Zones 2015 dataset (https://doi.org/10.7910/DVN/KJFUO1). Crop harvesting dates with a resolution of 0.5°×0.5° were provided by the Center for Sustainability and the Global Environment (Sacks et al., 2010). For crops having more than one growing season in a year, only the primary growing period was considered.

**2.2 GEOS-Chem model**

We used the GEOS-Chem global 3-D chemical transport model version 12.2.0 (http://acmg.seas.harvard.edu/geos/), driven by assimilated meteorological data from Modern Era Retrospective-analysis for Research and Applications, Version 2 (MERRA2) (https://gmao.gsfc.nasa.gov/reanalysis/MERRA-2/) with a horizontal resolution of 2.0° latitude by 2.5°



longitude and reduced vertical resolution of 47 levels. GEOS-Chem incorporates meteorological conditions, emissions,
chemical information, and surface conditions to simulate the formation, transport, mixing and deposition of ambient $O_3$. It
performs fully coupled simulations of $O_3$-$NO_x$-VOC-aerosol chemistry (Bey et al., 2001). Previous studies have
demonstrated the ability of GEOS-Chem to reasonably reproduce the magnitudes and seasonal variations of surface $O_3$
East Asia (Wang et al., 2011; He et al., 2012). To provide long-term simulated $O_3$ fields for incorporation into the ML
model (see below), we conducted GEOS-Chem simulations at a resolution of 2.0°×2.5°; higher resolutions of GEOS-Chem
in nested grids are available but computationally prohibitive for multi-decadal simulations. The original unit of GEOS-
Chem-simulated $O_3$ is ppb, which was converted to $\mu g\ m^{-3}$ assuming a constant temperature of 25°C and pressure of
1013.25 hPa (1 $\mu g\ m^{-3}$ is approximately 0.5 ppb) when compared with observations (Yin et al., 2017b; Gong and Liao,
2019).

Global anthropogenic emissions of CO, $NO_x$, $SO_2$ and VOCs are from Community Emissions Data System (CEDS),
which has coverage over the simulation years of 1950–2014 (Hoesly et al., 2018). Biomass burning emissions are from the
GFED-4 inventory (Van Der Werf et al., 2017). Biogenic VOC emissions are computed by the Model of Emissions of
Gases and Aerosols from Nature (MEGAN) v2.1 (Guenther et al., 2012), which is embedded in GEOS-Chem. Emissions
of biogenic VOC species in each grid cell, including isoprene, monoterpenes, methyl butenol, sesquiterpenes, acetone and
various alkenes, are simulated as a function of canopy-scale emission factors modulated by environmental activity factors
to account for changing temperature, light, leaf age, leaf area index (LAI), soil moisture and $CO_2$ concentrations
(Sindelarova et al., 2014).
Dry deposition follows the resistance-in-series scheme of Wesely (1989), which depends on species properties, land
cover types and meteorological conditions, and uses the Olson land cover classes with 76 land types reclassified into 11
land types. Although transpiration is a potential mechanism via which the land cover affects ozone, we do not address it in
this study because water vapor concentration in GEOS-Chem is prescribed from assimilated relative humidity (i.e., not
computed online from evapotranspiration).
**2.3 LightGBM machine-learning model**
In this study, we used the LightGBM algorithm to integrate GEOS-Chem simulated $O_3$ at a lower resolution with
higher-resolution multi-source data to produce higher-resolution hourly $O_3$ and MDA8-$O_3$ fields. Because the
representation of input data for LightGBM should be regular, datasets at different spatial resolutions were all regridded to
a unified resolution of 0.25°×0.25°, consistent with the meteorological fields. By taking the advantage of these high-
resolution datasets, the hybrid approach can not only correct the biases of the GEOS-Chem-simulated $O_3$, but also refine it
into a finer resolution. LightGBM is a ML algorithm based on the gradient boosting decision tree (Chen and Guestrin,
2016), which has a high training efficiency and lower memory footprint, and thus is suitable for processing massive high-
dimensional data (Zhang et al., 2019). The general steps to build a ML model can be summarized as follows: (1) choose
an algorithm appropriate for the problem (e.g., regression or classification); (2) clean the data and split them into training
and test data; (3) train and tune the model with training data to well capture prediction patterns; (4) evaluate model
performance on test data; and (5) return to step (3) and (4) until an optimal predictive ability is reached. The whole dataset
is divided into training and test data to evaluate the model generalization ability. The model performance on test data can
indicate whether the model can perform well on new data independent of the training process. A timescale of a year has
been suggested to strike a good balance between computational burden and utility for air quality forecasting, as the
variability in the power spectrum of surface $O_3$ can be captured by timescales of a year or less (Ma et al., 2021). Thus, in
this study, data for 2016–2017 were used as the training data, and data for 2018 were used as the independent test data. In
any process involving comparison with $O_3$ observations at site, the data or results from the nearest grid cells were used.
During the model training process, the model was evaluated with 10-fold cross-validation to ensure the robustness
and reliability of the model, whereby the training data were randomly partitioned into 10 subsets of approximately the same



size, with 90% of data used to train individual models and the ensemble model, and the remaining 10% of data used to
examine model performance (Xiao et al., 2018). This process was repeated 10 times so that each data record was left for
testing once. The tuning of the hyperparameters was optimized using grid search optimization to improve detection
performance and diagnostic accuracy (Wang et al., 2019). Statistical indicators, including the coefficient of determination
($R^2$) and root-mean-square error (RMSE), were used in subsequent assessment of model performance for GEOS-Chem
alone and for the hybrid approach.
**2.4 Ozone exposure metric and dose–response functions**
Among $O_3$ exposure indices, AOT40 has been used widely during the last two decades as it has been found to have a
strong relationship with relative yield of many crop species (Mills et al., 2007), and thus was used to quantify the impacts
of surface $O_3$ on crop yields in this study. The flux-based metrics, which require long-term simulations using a process-
based stomatal uptake model, were beyond the scope of this study. The AOT40 (ppm-h) is defined as follows:
$$\text{AOT40} = \sum_{i=1}^{n}([O_3]_i - 0.04) \tag{1}$$
where the $[O_3]_i$ is the hourly mean $O_3$ concentration (ppm) during the 12 hours of local daytime (08:00–19:59); $n$ is the
number of hours in the growing season defined as the 90 days prior to the start of the harvesting period according to the
crop calendar.
The exposure–response functions based on extensive field experimental studies have been established to relate a
quantifiable $O_3$-exposure metrics to crop yields. It has been suggested that , suggesting greater RYL responses found in
Asian experiments than the American and European counterparts, and possibly higher $O_3$ sensitivity of Asian crop varieties
(Emberson et al., 2009; Feng et al., 2022).To better understand $O_3$-induced risks to crops in China, the AOT40 dose-yield
functions developed based on field experiments in China are used in this study, which are named as AOT40-China. The
dose–response functions for soybean is from Zhang et al. (2017), and for other three crops are from Feng et al. (2022). The
statistical dose-yield relationships used in this study are summarized in **Table S1**.
**2.5 Analysis of health impacts**
All-cause mortality, cardiovascular disease mortality and respiratory disease mortality are selected as the health
outcomes of our study due to the high correlation between these endpoints and short-term $O_3$ exposure in previous studies.
A log-linear exposure-response function is widely adopted and recommended by the World Health Organization (WHO)
for health impact assessment in areas with severe air pollution. In particular, the log-linear model is the most widely applied
exposure-response model at present in China (Lelieveld et al., 2015; Yin et al., 2017a; Zhang et al., 2022b). The premature
mortality is calculated following:
$$\Delta M = \delta c * \left[ \frac{(\text{RR} - 1)}{\text{RR}} \right] * P \tag{2}$$
where $\Delta M$ is the excess mortality attributable to $O_3$ exposure; $\delta c$ is the baseline mortality rate for a particular health endpoint
(Yin et al., 2017b; Madaniyazi et al., 2016); $P$ is the exposed population; and RR is the relative risk defined as:
$$\text{RR} = \exp(\ (X - X_0) * \beta) \tag{3}$$
where $\beta$ is the exposure-response coefficient derived from epidemiological cohort studies (Shang et al., 2013); $X$ represents
the model-calculated $O_3$ concentration; the value of $X_0$ is the threshold concentration below which no additional risk is
assumed. Consistent with previous studies (Lelieveld et al., 2015; Liu et al., 2018), we used $X_0 = 75.2\ \mu g\ m^{-3}$.
In this study, the mean MDA8-$O_3$ concentrations in warm season (May-September) were used to estimate the disease-
specific health impacts of short-term exposure to $O_3$. The province-level population and national baseline mortality rate for





particular diseases were provided by the National Bureau of Statistics ([http://www.stats.gov.cn/](http://www.stats.gov.cn/)). The spatial differences of
baseline mortality in China were not considered without provincial-level data, which means that we assume the baseline
mortality is evenly distributed across China (Dedoussi et al., 2020). The exposure-response coefficients were obtained from
existing epidemiological studies in China (**Table S2**). If the corresponding coefficient of a province could not be found in
published epidemiological studies, the datum closest to that province would be selected as a substitute. If there were no
neighboring provinces, the results of national meta-analysis would be used (Zhang et al., 2021).
**3   Results**
**3.1 Model development and validation**
The finally selected features and their importance estimated by the LightGBM algorithm based on 10-fold cross
validation are shown in **Fig. 2**. GEOS-Chem-simulated $O_3$ is the top predictor for predicting surface $O_3$ concentrations,
accounting for 61% and 58% of all relative importance in the ML algorithm predicting hourly $O_3$ and daily MD8A-$O_3$,
respectively. The result indicates that process-based GEOS-Chem simulations have high utility for $O_3$ predictions under
the hybrid approach (Ma et al., 2021). The meteorological variables with high contribution to both the daily and hourly
models are downward surface solar radiation (SSRD), relative humidity at 1000 hpa (RH_1000hpa) and 10-m horizontal
wind (U10 and V10). Other special features, including location (latitude and longitude), elevation and diurnal and monthly
pattern of $O_3$, also contribute to ambient $O_3$ estimations. The spatial distributions of bias-corrected $O_3$ are consistent with
observations for both training and test datasets (**Fig. S1**), indicating that there is no obvious overfitting, i.e., the model is
able to generalize from the training set to the test set. The good generalization ability of the model gives us confidence in
its ability to make accurate predictions based on new data. In general, the hybrid approach can yield good $O_3$ estimates in
the data-intensive regions, including eastern and central China that are the hotspot areas of $O_3$ pollution.

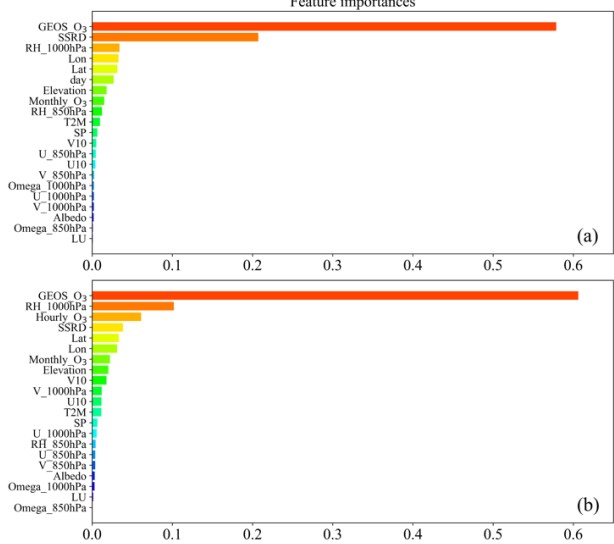

**Figure 2. The feature importance plot for (a) MDA8-$O_3$ and (b) hourly $O_3$, respectively. The full list of candidate**
**variables with their symbols, units, descriptions, and data sources are shown in Table S1.**

**Fig. 3** shows the density scatter plots between $O_3$ measurements and GEOS-Chem simulations, as well as the hybrid-
approach predictions for 2018. The $R^2$ value of the hybrid approach and GEOS-Chem model are 0.66 and 0.27 at hourly
level, and 0.72 and 0.53 at MDA8-$O_3$ level, respectively. Bias-corrected $O_3$ concentrations have lower RMSE in
comparison with GEOS-Chem simulated $O_3$ concentrations, reduced from 31.1 to 23.8 μg m$^{-3}$ for MDA8-$O_3$ predictions,
and from 38.5 to 26.3 μg m$^{-3}$ for hourly predictions. The MDA8-$O_3$ model performance is better than that of the hourly
model, indicating reduced errors upon temporal averaging. The result suggests that the CTM-simulated results can be
substantially improved by applying ML with multi-source datasets, and the bias-corrected data can improve our
understanding of long-term $O_3$ trends and its further implications on crop and human health over China, as discussed in the
following sections.
In comparison with previous studies, Liu et al. (2020) used XGBoost to predict $O_3$ in major urban areas of China at a
resolution of 0.1°×0.1°, and the $R^2$ value and RMSE for MDA8-$O_3$ were 0.74 and 23.8 μg m$^{-3}$, respectively. Their result
indicates that higher-resolution predictions may help enhance model accuracy, but represent a trade-off between model
accuracy and time efficiency depending on the purpose. Instead of directly predicting $O_3$ concentrations, Ivatt and Evans
(2020) predicted biases in GEOS-Chem-simulated $O_3$ concentrations and then corrected them with XGBoost. They also
suggested that the corrected model performs considerably better than the uncorrected model, with RMSE reduced from
16.2 to 7.5 ppb and Pearson's $R$ raised from 0.48 to 0.84. Their greater improvement with larger reduced RMSE than our
result is mainly because they selected fewer sites for training, with all the urban and mountain sites (observations made at
a pressure < 850 hPa) removed. The removal of these sites can improve the overall apparent performance of the model
because $O_3$ formation could have different characteristics in these areas. In general, ML methods have been proven to be a
promising tool to improve air pollutant forecasts when a process-level understanding is still incomplete.

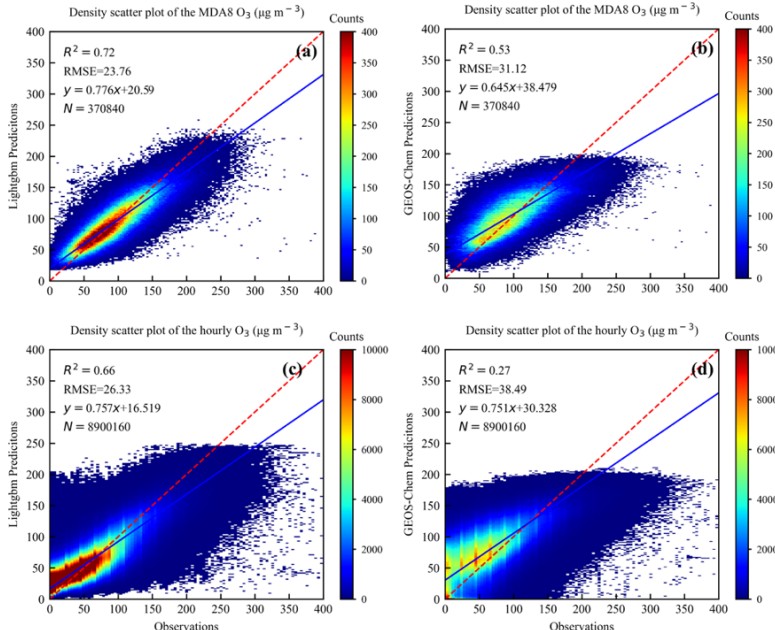


**Figure 3. Density scatter plots and linear regression statistics of $O_3$ predictions vs. observation for 2018: (a) bias-**
**corrected MDA8-$O_3$ vs. observations; (b) GEOS-Chem MDA8-$O_3$ vs. observations; (c) bias-corrected hourly $O_3$ vs.**
**observations; and (d) GEOS-Chem hourly $O_3$ vs. observations. The dashed red line indicates the 1:1 line, and the**
**solid blue line indicates the line of best fit using orthogonal regression. The $R^2$ is the coefficient of determination,**





**RMSE is the root-mean-square error, and *N* is the number of data points. The X and Y axis represents the O₃**
**observations and predictions, respectively.**

**3.2  Spatiotemporal distribution and trends of O₃ predictions**
**Fig. 4** demonstrates the spatial patterns of averaged annual and warm-season (May-September) MDA8-O₃ from 1981
to 2019. When compared to the high concentrations in the warm season, MDA8-O₃ concentrations are relatively lower at
annual level. The annual and warm-season MDA8-O₃ concentrations have similar spatial distribution, and both present an
increasing trend over the past decades, with more substantial increase observed between 1981 and 2010. The O₃ levels in
southern China are lower than those in northern China, but they are still relatively high in the PRD region, which is
consistent to findings in previous studies (e.g. Liu and Wang, 2020b). During the first decade of 1981–1990, high O₃
concentration areas are mainly concentrated in the BTHs and northern Shandong. In the next two decades, O₃ pollution
extensively expands to most of East and North China, spreading northward to Jilin and Liaoning, westward to Shanxi and
Ningxia, and southward to northern Hunan, Shanxi and Zhejiang. Moreover, the SCB and PRD regions also experience
aggravated O₃ pollution during this period. In the last decade of the study period, O₃ concentrations remain at high levels
in BTHs and SCB without obvious changes. To understand the detailed changes and trends of O₃, next we analyze the
interannual variability.

## MDA8 O₃ (μg m⁻³)

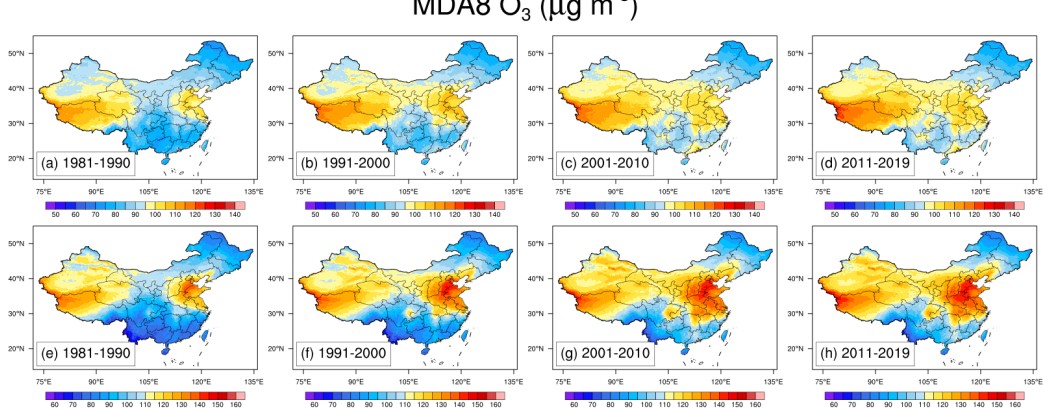

**Figure 4. Spatial distribution of the annual mean MDA8-O₃ concentrations (μg m⁻³) during: (a) 1981–1990; (b)**
**1991–2000; (c) 2001–2010; and (d) 2011–2019. Spatial distribution of the warm-season (May-September) mean**
**MDA8-O₃ concentrations of (e)1981–1990, (f) 1991–2000, (g) 2001–2010; and (h) 2011–2019.**
**Fig. 5** shows that the annual averaged MDA8-O₃ concentrations increase from 87 μg m⁻³ in 1981 to 98 μg m⁻³ in
2019, with a growth rate of +0.26 μg m⁻³ yr⁻¹, while the warm-season averaged MDA8-O₃ concentrations increase from
100 μg m⁻³ in 1981 to 117 μg m⁻³ in 2019, having a growth rate of +0.51 μg m⁻³ yr⁻¹. Moreover, the average annual and
warm-season O₃ concentrations have a more obvious upward trend before 2000s, with a growth rate of 0.38 μg m⁻³ yr⁻¹
and 0.71 μg m⁻³ yr⁻¹, compared to that after 2000s, when O₃ concentrations appear to fluctuate within a certain range.
GEOS-Chem-simulated O₃ has a similar trend as the bias-corrected O₃, but it generally overestimates O₃ concentrations on
national scale (**Fig. S2**). The annual and warm-season averaged MDA8-O₃ concentrations in BTHs, YRD, SCB and PRD
regions are shown in **Fig. S3–S4**. The warm-season increasing trend for BTHs, YRD, SCB and PRD regions are 0.32 μg
m⁻³ yr⁻¹, 0.63 μg m⁻³ yr⁻¹, 0.84 μg m⁻³ yr⁻¹, and 0.81 μg m⁻³ yr⁻¹ from the year 1981 to 2019.

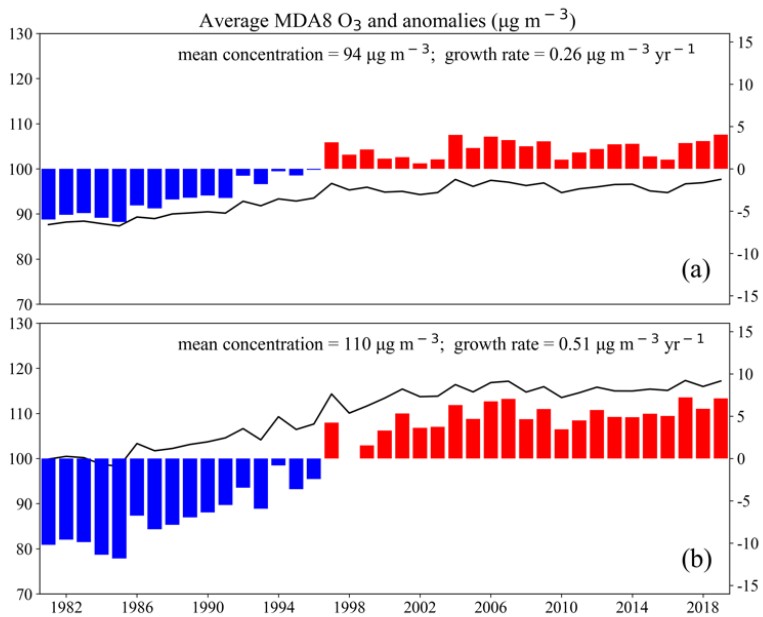


**Figure 5. The bias-corrected MDA8-O$_3$ predictions (black line; left y axis) and corresponding anomalies (colored**
**bar; right y axis) from 1981 to 2019: (a) annual mean; and (b) warm-season mean (May-September). The trends**
**(growth rates) are obtained by ordinary linear regression on mean values of MDA8-O$_3$. The anomalies are defined**
**as annual mean minus the multidecadal average over 1981–2019.**

In recent years, the worsening O$_3$ pollution has fueled numerous studies on ground-level O$_3$ spatial distribution and
changes in China, which were conducted on local, regional and national scale using different O$_3$ fields from observations,
CTMs and ML estimates. In this study, we mainly focus on the regional and national O$_3$ characteristics, and the reported
O$_3$ trends in recent studies are listed in **Table 1**. By comparing the results of existing works, we find that source-varied O$_3$
fields can induce great uncertainty of the O$_3$ trends. Moreover, the O$_3$ trends are found to be very sensitive to the study
period even with the same O$_3$ fields (Wei et al., 2022), which indicates large interannual variability, mostly reflecting the
changing anthropogenic emissions and meteorology (Lu et al., 2019; Li et al., 2020). In contrast to the perceptible O$_3$ trends,
Liu et al. (2020) suggested that O$_3$ pollution in most parts of China has only modest changes between 2005 and 2017, and
their trends were not spatially continuous. Wang et al. (2022b) also reported that O$_3$ has small positive increase rates for
2013–2021 in many cities, and the O$_3$ increase rates greatly differ from site to site even within the same region.
In comparison, our results indicate no obvious increasing trends of national MDA8-O$_3$ within the same study period
(**Fig. 5**). On a regional scale, only BTHs have a perceptible increasing trend in more recent years, while no such trends are
found over the YRD, SCB and PRD regions during the same period. The summertime MDA8-O$_3$ in BTHs has a change
rate of +0.81 $\mu g\ m^{-3}\ yr^{-1}$, which is much lower than the results using O$_3$ observations (Li et al., 2020). One possible reason
is that most observational sites are in urban regions, which usually suffer more serious O$_3$ pollution, while the O$_3$
concentrations from model simulations and ML methods are calculated on the scale of a grid cell with lower domain-
averaged values. Moreover, gridded data at a relatively coarse resolution may fail to capture larger site differences, leading
to the larger discrepancy of between O$_3$ observations and gridded O$_3$ estimates.





**Table 1 Summary of reported regional and national MDA8-O$_3$ trends (µg m$^{-3}$ yr$^{-1}$).**

| Region | Period | Increase rate | Data source/Method | References |
|---|---|---|---|---|
| Nation | 2013–2017 (annual) | 0.35 | ML (XGBoost) | (Liu et al., 2020) |
| | 2013–2017 (annual) | 0.92 | WRF-CMAQ | (Liu and Wang, 2020a) |
| | 2013–2017 (annual) | 1.33 | ML (ERT) | (Wei et al., 2022) |
| | 2015–2019 (annual) | 4.40 | ML (ERT) | (Wei et al., 2022) |
| | 2015–2019 (annual) | 1.90 | Observations | (Maji and Namdeo, 2021) |
| | 2013–2019 (summer) | 3.80 | Observations | (Li et al., 2020) |
| | 1981–2019 (annual) | 0.26 | ML (LightGBM) | This study |
| | 1981–2000 (annual) | 0.38 | ML (LightGBM) | This study |
| | 1981–2019 (warm-season) | 0.51 | ML (LightGBM) | This study |
| | 1981–2000 (warm-season) | 0.71 | ML (LightGBM) | This study |
| BTH | 2010–2017 (annual) | 0.60 | ML (Random Forest) | (Ma et al., 2021) |
| | 2013–2017 (annual) | 1.33 | ML (XGBoost) | (Liu et al., 2020) |
| | 2013–2017 (annual) | 4.78 | ML (ERT) | (Wei et al., 2022) |
| | 2012–2017 (summer) | 1.16 | GEOS-Chem | (Dang et al., 2021) |
| | 2013–2019 (summer) | 6.60 | Observations | (Li et al., 2020) |
| | 1981–2019 (summer) | 0.46 | ML (LightGBM) | This study |
| | 2013–2019 (summer) | 0.81 | ML (LightGBM) | This study |
| YRD | 2013–2017 (annual) | 2.94 | ML (ERT) | (Wei et al., 2022) |
| | 2015–2019 (annual) | 5.60 | ML (ERT) | (Wei et al., 2022) |
| | 2012–2017 (summer) | 3.48 | GEOS-Chem | (Dang et al., 2021) |
| | 2013–2019 (summer) | 3.20 | Observations | (Li et al., 2020) |
| | 1981–2019 (annual) | 0.24 | ML (LightGBM) | This study |
| | 1981–2019 (summer) | 0.73 | ML (LightGBM) | This study |
| SCB | 2013–2017 (annual) | 2.37 | ML (ERT) | (Wei et al., 2022) |
| | 2013–2019 (summer) | 1.40 | Observations | (Li et al., 2020) |
| | 1981–2019 (annual) | 0.48 | ML (LightGBM) | This study |
| | 1981–2019 (summer) | 0.98 | ML (LightGBM) | This study |
| PRD | 2007–2017 (annual) | 1.20 | Observations | (Yang et al., 2019) |
| | 2013–2017 (annual) | −0.72 | ML (ERT) | (Wei et al., 2022) |
| | 2015–2019 (annual) | 4.38 | ML (ERT) | (Wei et al., 2022) |
| | 2013–2019 (summer) | 2.20 | Observations | (Li et al., 2020) |
| | 1981–2019 (annual) | 0.56 | ML (LightGBM) | This study |
| | 1981–2019 (fall) | 0.69 | ML (LightGBM) | This study |


**3.3 Seasonal characteristics of O$_3$ predictions**
Differences in averaged annual and warm-season O$_3$ concentrations indicate that O$_3$ has distinctive seasonal
characteristics. **Fig. 6a-d** shows the seasonal variations in O$_3$ concentrations from 2011–2019, and results for other past
three decades are shown in **Fig. S5-S7**. In winter, pollution is mainly concentrated in the coastal areas of southern China.
In spring, O$_3$ pollution primarily occurs in eastern China and the southern part of Yunnan Province. O$_3$ pollution continues
to aggravate over eastern China in summer, particularly in BTHs, and further extends to SCB. The air quality in eastern
and central China is greatly improved in fall, while southern China experiences the most pollution in this period. In general,
the peak and trough values of O$_3$ concentrations appear in summer and winter, respectively. However, O$_3$ concentrations
are found to be minimum in summer and maximum in fall over PRD, which is largely determined by the summer monsoon
(Zhou et al., 2013; Wang et al., 2018). **Fig. S8** shows the seasonal averaged MDA8-O$_3$ concentrations in different regions
from 1981 to 2019. In winter, O$_3$ concentrations do not have much change across the four regions over the past decades,
staying mostly between 70–80 µg m$^{-3}$. Moreover, wintertime O$_3$ concentrations after the 2000s are generally lower than
that before the 2000s in BTHs, YRD and SCB. In contrast, summertime O$_3$ concentrations have a dramatic increase over
the four regions. In spring and fall, O$_3$ concentrations have an increasing trend in PRD, while it mostly fluctuates within a



certain range in the other three regions. The results show that O₃ in non-winter seasons has a more pronounced increase
during 1981–2019 albeit with regional differences. The regional characteristics of O₃ and its influencing factors will be
further discussed in Section 3.4.

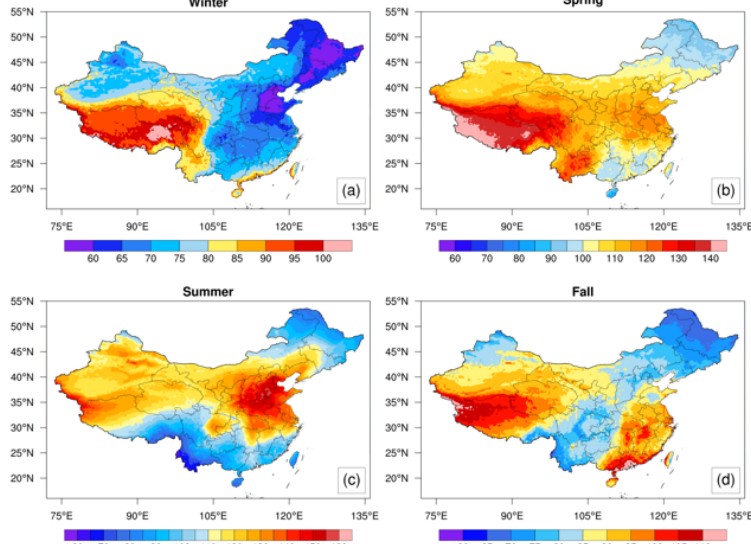

**Figure 6. Spatial distribution of the bias-corrected MDA8-O₃ predictions (µg m⁻³) from 2011–2019: (a) winter; (b)**
**spring; (c) summer; and (d) fall.**

**3.4 Regional characteristics of O₃ predictions**
**Fig. 7** shows the bar plots of the seasonal MDA8-O₃ concentrations in each region from 1981–2019 for bias-corrected
and GEOS-Chem-simulated O₃. For the bias-corrected O₃, the averaged summertime MDA8-O₃ concentrations in BTHs,
YRD, SCB and fall-time MDA8-O₃ concentrations in PRD are $137 \pm 8$ µg m⁻³, $119 \pm 10$ µg m⁻³, $113 \pm 12$ µg m⁻³ and $98 \pm$
$10$ µg m⁻³, with the increasing rate being $0.46$ µg m⁻³ yr⁻¹, $0.73$ µg m⁻³ yr⁻¹, $0.98$ µg m⁻³ yr⁻¹ and $0.69$ µg m⁻³ yr⁻¹ from 1981
to 2019, respectively (**Fig. S9**). For GEOS-Chem-simulated O₃, the averaged summertime MDA8-O₃ concentrations in
BTHs, YRD, SCB and fall-time MDA8-O₃ concentrations in PRD are $141 \pm 7$ µg m⁻³, $125 \pm 11$ µg m³, $120 \pm 14$ µg m⁻³
and $100 \pm 12$ µg m⁻³, respectively. It shows that O₃ concentrations of the four regions have a consistent upward trend in
the summer over the past decades, but there are regional differences in other seasons. Compared to BTHs and YRD, PRD
and SCB have more distinctive O₃ increases in spring and fall. Among these four regions, the O₃ concentrations in BTHs
has the biggest seasonal differences, but have the smallest seasonal differences in PRD.
The spatiotemporal patterns of O₃ in China have been proven to largely depend on both emissions and meteorology.
The regional O₃ pollution is usually found to be triggered by specific circulation patterns as local meteorological factors
are modulated by synoptic-scale circulation patterns. China has a large territory and is affected by different weather systems.
The continental high-pressure systems, components of East Asian summer monsoon (EASM) and tropical cyclones, among
others, are critical synoptic conditions leading to O₃ formation and transport in China (Wang et al., 2022b; Han et al., 2020).
For instance, regional O₃ pollution in North China usually occurs under a typical weather pattern of an anomalous high-
pressure system at 500 hPa (Gong and Liao, 2019), which creates favorable meteorological conditions for high O₃ levels
with high temperature, low relative humidity, anomalous southerlies and divergence in the lower troposphere. As one of
the most important components of EASM, the Western Pacific subtropical high (WPSH) strongly influences summertime
precipitation and atmospheric conditions in East China. A strong WPSH can decrease O₃ levels over YRD as enhanced



moisture is transported into YRD under prevailing southwesterly winds (Zhao and Wang, 2017). Located on the southern coast of China, PRD features a typical subtropical monsoon climate. There O₃ concentrations are usually the lowest in summer due to the prevailing southerlies with clean air from the ocean and the associated large rainfall, while the worst O₃ pollution usually happens in fall mainly due to the occasional northerly winds during the monsoonal transition, thereby importing precursors from the north, and stable and still relatively warm and sunny weather conditions before the winter starts. Downdrafts in the periphery circulation of a typhoon system can also strongly enhance surface O₃ before typhoon landing (Jiang et al., 2015; Lu et al., 2021; Li et al., 2022). On one the hand, the poor ventilation in the peripheral subsidence region of typhoons favors the accumulation of O₃ and its precursors. On the other hand, the deep subsidence can transport the O₃ in the upper troposphere and lower stratosphere to surface, causing aggravated O₃ pollution. Moreover, smaller-scale circulation patterns, such as land-sea and mountain-valley breezes, also influence O₃ in coastal regions (Ding et al., 2004; Zhou et al., 2013; Wang et al., 2018).

When compared to the hybrid approach, GEOS-Chem generally has similar O₃ distribution and trends over each region, while overestimating O₃ concentrations (Table S1). GEOS-Chem particularly overestimates wintertime and fall-time O₃ concentrations in SCB, which are $10 \pm 1$ µg m⁻³ and $17 \pm 3$ µg m⁻³ higher than those of the hybrid approach, respectively. Previous studies reported such model overestimates and proposed a number of explanations involving precursor emissions, dry deposition, and vertical mixing in the planetary boundary layer (PBL), etc. Both observational analyses and inter-model comparisons suggested that the summertime dry deposition of O₃ calculated by the Wesely scheme in GEOS-Chem could be underestimated, which has been invoked as a cause for model overestimates of O₃. The biased emissions in the model, as consistent with the biased-high tropospheric NO$_x$ columns, result in overestimated O₃. Travis et al. (2016) showed that GEOS-Chem with reduced NO$_x$ emissions provides an unbiased simulation of O₃ observations from the aircraft and reproduces the observed O₃ production efficiency in the boundary layer. Lin et al. (2008) suggested that the excessive PBL mixing can lead to the biased-high O₃ concentrations. The fully mixed O₃ throughout the PBL means that the higher O₃ concentrations in the upper PBL are brought down to the surface much more efficiently. Moreover, the excessive spatial averaging of emissions at coarser resolutions could also lead to systematic overestimation of regional O₃ production (Wang et al., 2013). In summary, with a higher prediction accuracy, the hybrid approach lends greater credence to using model simulations to extrapolate historical O₃ further back in time, which can furthermore provide us with more accurate estimates of O₃ impacts on crop production and human health.

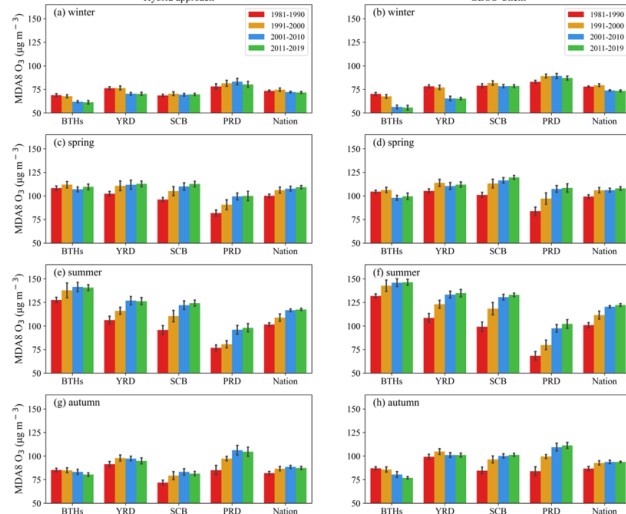

**Figure 7. The seasonal mean MDA8-O₃ concentrations (µg m⁻³) in different regions during 1981-2019. Bias-**



corrected MDA8-O$_3$ in: (a) winter; (c) spring; (e) summer; and (g) fall. GEOS-Chem MDA8-O$_3$ in: (b) winter; (d)
**spring; (f) summer; and (h) fall. The error bar represents the standard deviation.**
**3.5 Crop production losses attributable to O$_3$ pollution**
**Fig. 8** shows the relative yield losses (RYLs; RYL = 1 – RY, where RY is the relative yield defined as the ratio of the
O$_3$-affected yield to the yield without O$_3$ exposure) calculated with GEOS-Chem and bias-corrected O$_3$ using AOT40-
China metric**.** For a given crop, the RYLs show generally consistent spatial distribution across the metrics, with BTHs
having the most serious crop yield losses due to high O$_3$ concentrations. Compared to the bias-corrected O$_3$, using GEOS-
Chem-simulated O$_3$ generally leads to larger yield losses, especially over BTHs and SCB, reflecting overestimated O$_3$
concentrations by GEOS-Chem in cropland areas during the growing seasons (**Fig. S11**), primarily in spring and summer,
which is consistent to the above analysis. GEOS-Chem-simulated O$_3$ leads to slightly underestimated wheat yield loss only
over some parts of BTHs, mostly because the primary growing period of wheat there is in winter and spring, and GEOS-
Chem has lower O$_3$ estimates than the hybrid approach during this period there (**Table S2**).

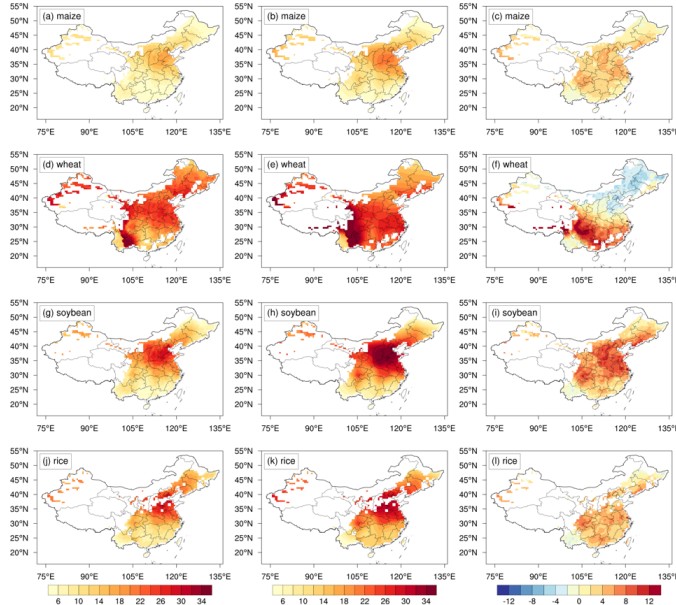


**Figure 8. Estimated annual mean relative yield losses (RYLs, in %) of four staple crops from 1981–2019 using the**
**AOT40-China metric. The estimated RYLs using bias-corrected O$_3$: (a) maize; (d) wheat; (g) soybean; and (j) rice.**
**The estimated RYLs using GEOS-Chem-simulated O$_3$: (b) maize; (e) wheat; (h) soybean; and (k) rice. The**
**differences in estimated RYLs between GEOS-Chem-simulated and bias-corrected O$_3$: (c) maize; (f) wheat; (i)**
**soybean; and (l) rice. The GEOS-Chem-simulated O$_3$ were regridded to 0.5°×0.5° for comparison with bias-**
**corrected O$_3$.**
**Fig. 9** shows the bar plots of the relative yield for each crop using AOT40-China dose-yield relationship. Crop yield
losses are generally consistent with the O$_3$ trends as the dose-yield relationships used here are essentially a set of linear
functions. Most crops experience aggravated yield losses over the past four decades due to enhanced O$_3$ concentrations,
except for wheat, which has the largest yield loss during the period 1991 to 2000. The reason could be that BTHs have the





highest O$_3$ concentrations in spring during the 1990s (**Fig. S10**), which is the primary growing season for wheat. Noticeable
uncertainties of crop yield losses are found across metrics.
The average annual crop RYLs from 1981 to 2019 for wheat, rice, soybean and maize range from 1.1 to 13.4%, 2.7 to
13.4%, 6.3 to 24.8% and 0.8 to 7.4%, respectively. The differences in yield losses across crops reflect the dependence on
crop-specific phenology and ecophysiology. The estimated annual RYLs using bias-corrected O$_3$ for wheat, rice, soybean
and maize from 1981 to 2019 range from 17.5–25.5%, 10.7–19.1%, 7.3–17.9% and 7.1–12.7%, with a growth rate of 0.03%
yr$^{-1}$, 0.04% yr$^{-1}$, 0.27% yr$^{-1}$ and 0.13% yr$^{-1}$. Wheat is the most sensitive crop to the O$_3$ concentrations, whereas maize is
the least sensitive. Using GEOS-Chem-simulated O$_3$, the estimated annual RYLs for wheat, rice, soybean and maize from
1981 to 2019 are 18.7–28.7%, 14.0–22.0%, 12.4–23.1%, and 7.9–13.2%, having a growth rate of 0.08% yr$^{-1}$, 0.14% yr$^{-1}$,
0.23% yr$^{-1}$ and 0.11% yr$^{-1}$. There are noticeable differences in crop yield estimates using the bias-corrected and GEOS-
Chem O$_3$, indicating again the importance of the bias-corrected high-resolution O$_3$ data in related crop issues.
In existing studies evaluating the O$_3$-induced crop losses in China, which also use dose-yield relationship derived from
the experiments conducted in Asia, Zhang et al. (2017) reported that the ambient O$_3$ concentrations in Northeast China
cause substantial annual yield loss of soybean ranging from 23.4% to 30.2% during 2013 and 2014, depending on the O$_3$
metric used (including AOT40, W126, SUM06 and a flux-based metric). Feng et al. (2022), using AOT40, indicated that
the annual average RYLs of wheat (33%), rice (23%) and maize (9%) from 2017 to 2019. Our correspondingly estimated
RYLs for rice (18.0%) and maize (10.0%) are generally consistent to their results, while the RLYs for soybean (16.4%)
and wheat (23.4%) are much lower than the estimates. Since we used the same dose-response relationships in their studies,
the discrepancies are primarily attributed to the differences in used metrics (only for soybean), O$_3$ fields and sensitivity of
crop to the changes of O$_3$ concentrations (Mukherjee et al., 2021; Feng et al., 2022; Mills et al., 2018). In Zhang et al.
(2017), the O$_3$ measurements are obtained from the experimental field (45°73′N, 126°61′E), and in Feng et al. (2022), the
measured O$_3$ concentrations are from over 3,000 monitoring sites across East Asia. The results of comparison are consistent
to the previous analysis of O$_3$ trends and variability from different sources, where the domain-average values of O$_3$
observations are larger than gridded O$_3$ from model simulations (**Section 3.2**) and thus lead to larger estimates of RYLs.
On one hand, it indicates that O$_3$ fields should be considered as a great source of uncertainty when comparing the results
of previous studies using source-varied O$_3$ fields. Moreover, different degrees of importance should be given for specific
crops, for example, the changes in O$_3$ concentrations have a larger impact on wheat crop. On the other hand, it highlights
again the necessity and importance of bias correction for model-simulated O$_3$ when O$_3$-inudec crop reduction.

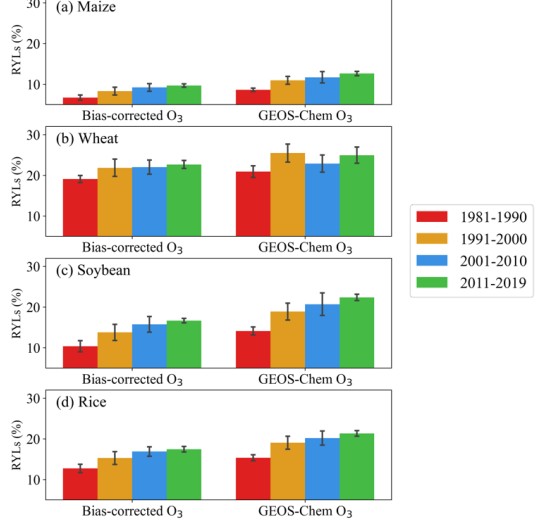




**Figure 9. The estimated decadal mean relative yield losses (RYLs) of four staple crops using different metrics from 1981–2019. The estimated RYLs using bias-corrected O₃: (a) maize; (c) wheat; (e) soybean; and (g) rice. The estimated RYLs using GEOS-Chem-simulated O₃: (b) maize; (d) wheat; (f) soybean; and (h) rice. The error bar represents the standard deviation.**

**3.6 Health impacts attributable to O₃ pollution**

The exposure-response coefficients for the short-term, acute health impacts of O₃ are shown in **Table S4**. The estimated annual all-cause premature deaths induced by O₃ increase from 55,876 in 1981 to 162,370 in 2019 with an increasing trend of +2,979 deaths yr⁻¹. The annual premature deaths related to respiratory and cardiovascular diseases are 34,155 and 40,323 in 1998, and 26,471 and 79,021 in 2019, having a rate of change of –546 and +1,773 deaths yr⁻¹ during 1998–2019, respectively (**Fig. 10a**). Among three types of health outcomes, only respiratory diseases experienced a decreasing trend in premature mortality, and the premature mortality is constantly below 40,000. The decreasing trend of the respiration-related mortality primarily results from the decreased annual baseline mortality rate over the past decades (**Fig. S12**). As the total respiratory-related deaths decreased over the past decades, respiratory O₃ deaths are decreasing even under aggravated O₃ pollution. Based on GEOS-Chem-simulated O₃, the corresponding estimated change rate for all-cause disease is +3,516 deaths yr⁻¹ from 50,384 in 1981 to 176,741 in 2019. Premature mortality induced by respiratory disease decreases from 37,822 in 1998 to 29,079 in 2019 with a change rate of –584 deaths yr⁻¹, while cardiovascular disease increases from 44,516 in 1998 to 85,980 in 2019 with a change rate of +1,977 deaths yr⁻¹ (**Fig. S13**). The result shows that using GEOS-Chem-simulated O₃ generally gives higher estimates of mortality than using the bias-corrected data. **Fig. 10b** shows the provincial annual average premature mortality of different health endpoints. The five provinces with the highest all-cause mortality are Jiangsu [14,510 (95% CI: 9,022–19,935)], Shandong [12,684 (95% CI:4,258–20,990)], Henan [12,290 (95% CI: 4,125–20,343)], Guangdong [9,268 (95% CI: 7,224–11,416)] and Hebei [8,276 (95% CI: 2,776–13,706), which are generally consistent with previous studies for China (Zhang et al., 2021; Zhang et al., 2022a). Similar distribution can be found for respiratory and cardiovascular diseases but with a different ranking order. Generally, those provinces in densely populated areas (**Fig. 10c**) with higher O₃ concentrations, such as BTHs, YRD and PRD, have higher health burdens. In contrast, the northeastern and southern China (excluding Guangdong) suffer the least life losses induced by O₃ exposure (**Fig. S14**).

When compared with estimates from previous studies, our estimates of are generally quite consistent with that given by Maji and Namdeo (2021), which reported that the short-term all-cause, cardiovascular and respiratory premature mortalities attributed to ambient O₃ exposure were 156,000, 73,500 and 28,600 in 2019. Based on O₃ observations in 334 Chinese cities, Zhang et al. (2021) suggested that the national all-cause, respiratory, cardiovascular mortalities attributable to O₃ are 270,000 to 390,000, 49,000 to 63,000, and 150,000 to 220,000 million across 2015–2018, which are much higher than most existing results. Since the methodological approaches are largely similar and we use the log-linear exposure-response function, we ascribe that the very high estimated mortalities are mainly due to concentration–response threshold $X_0$ assumed to be zero in their study. A lower $X_0$ means that O₃ can cause more adverse impacts on human health even at low concentrations, thus leading to higher mortalities.



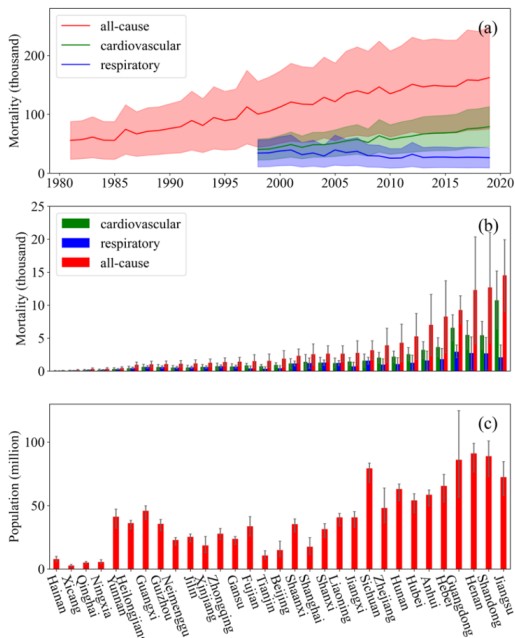

**Figure 10. (a) Annual premature morality (thousand) for different diseases over the past decades; (b) annual mean province-based morality (thousand) attributed to different health endpoints; and (c) annual mean province-based population (million). The morality is calculated using the bias-corrected O₃.**

### 4. Conclusions and discussion

In this study, to have a more accurate characterization of O₃ spatiotemporal distribution and trends as well as their impacts on agriculture and human health, we used a hybrid approach to generate bias-corrected O₃ data across China from 1981 to 2019. The hybrid approach helps improve O₃ predictions by taking advantage of a chemical transport model, a ML algorithm and increasing availability of high-resolution environmental and meteorological data. The validation shows that the bias-corrected O₃ can achieve a higher prediction accuracy than GEOS-Chem-simulated O₃ alone when compared with historical in-situ measurements. Before being corrected, the GEOS-Chem-simulated O₃ concentrations tend to be overestimated and lead to higher crop yield losses and larger O₃-induced mortalities. Noticeable differences in crop RYLs and mortality estimates highlight the advantages of using high-resolution O₃ data to improve our understanding of long-term O₃ impacts.

When examining the regional and national O₃ trends, we found that MDA8-O₃ concentrations have a perceptible increasing trend before 2000s, but fluctuate within a certain range with large interannual variabilities in more recent years. The large discrepancies in previous studies indicate that the regional and national O₃ trends in China still suffer with great uncertainties, particularly when different approaches are used to produce the O₃ estimates. However, these studies using source-varied O₃ fields consistently show the great interannual variabilities of O₃ concentrations. Some insights can be obtained from existing findings, which need to be carefully considered when examining O₃ trends and comparing them with existing results. First, given the large site differences, the calculation of observational O₃ trends is very sensitive to the subsets of data from networks. Thus, great uncertainty could still exist even using O₃ observations from the same source depending on the chosen subsets of data. Second, different formats of O₃ fields (e.g., site-based and gridded) could lead to



large uncertainties of the O₃ trend estimates. A higher resolution of gridded O₃ estimates from CTMs and ML may reduce
the differences between O₃ observational results. Third, the calculated O₃ trends are very sensitive to the chosen study
period due to large interannual variability and seasonal differences. The changing meteorological conditions are the major
factor causing the large interannual O₃ variations, and reductions in the emissions of NOₓ, SO₂ and PM also have complex
effects on ground-level O₃ concentrations (Wang et al., 2022b). Liu and Wang (2020b) suggested that the meteorological
impacts on O₃ trends vary region by region and year by year and could be comparable with or even larger than the impacts
of changes in anthropogenic emissions.
Our estimated RYLs for maize and rice and soybean in China are generally consistent to existing studies, while the
RLYs for soybean and wheat are lower than their estimates mainly due to the differences in used metrics, O₃ fields and
crop sensitivity to ambient O₃ concentrations. It suggests that plating O₃-resistant cultivars could be an effective approach
to increase total crop production to meet the increasing food demands. In addition to the metrics and O₃ fields, uncertainties
of estimated O₃-induced crop losses could be also from other sources (e.g., dose-yield relationships). Though some other
metrics (e.g., M7/M12 and W126) have also been used in some studies (Van Dingenen et al., 2009; Avnery et al., 2013;
Wang et al., 2022c), there are not available dose-relationships for all four major crops specific for China. The estimated
RYLs for crops could be largely biased using metrics with dose-yield relationships developed for U.S. or Europe (**Fig.S15**),
as they are inadequate to represent Asian crop genotypes and environmental conditions. So, the region-specific dose-yield
relationships are highly recommended to be used in future study estimating the O₃-induced crop reduction, especially for
the regional study. Moreover, it is worth noting that as the concentration-based metrics do not account for how crop
physiological responses to the changing atmospheric environment, the associated dose-yield relationships which is
currently useful may not hold in the future (Tai et al., 2021). So, the flux-based metrics and the process-based crop model
are more recommended to be used for future O₃ risk assessments, wherein more crop- and region-specific experiments and
trials are needed to acquire appropriate metrics and dose-response functions and calibrate the process-based crop model.
In recent years, although existing studies have made efforts to quantify the O₃-related health impacts in China, only a
few focused on the nationwide acute O₃ health burden assessment, particularly for assessment over multiple decades (Maji
and Namdeo, 2021; Sahu et al., 2021; Zhang et al., 2021; Zhang et al., 2022a). There are some remaining issues to be
addressed regarding O₃ health impacts. For instance, the existence of a "safe" threshold of O₃ levels still remains debated.
A recent study reported that no consistent evidence was found for a threshold in the O₃-mortality concentration-response
relationship in seven cities of Jiangsu Province, China during 2013–2014 (Chen et al., 2017; Maji and Namdeo, 2021).
Given the importance of the threshold assumption in assessing health effects of air pollution, more studies are needed to
determine a most likely threshold for O₃-mortality association in the future. Moreover, the multiple temporal O₃ metrics
(e.g.,1-h maximum and daytime average O₃ concentrations) have also been proved to play an important role in the
variability of estimated health effects, even though standard ratios are used to convert among multiple metrics (Anderson
and Bell, 2010). In addition to the uncertainties from varying methodologies, interpretation of the O₃ epidemiological
impact is also constrained by the variability in geographical, seasonal, and demographic characteristics (Yin et al., 2017b).
Liu et al. (2013) suggested that associations between O₃ and mortality appeared to be more evident during the cool season
than in the warm season, and stronger in the oldest age group and among those with less education. The effect modification
by population susceptibility and the confounding effects of concomitant exposures (temperature, particulate matter, etc.)
should be further considered in future works.
A major limitation of our study lies in the uncertain predictions in regions where monitoring data are scarce (e.g., the
western half of China). The monitoring sites are sparsely distributed in those areas, which may fail to capture the accurate
association between O₃ concentrations and various predictors there, especially considering that the ML algorithm has likely
over-emphasized such relationships in the data-intensive eastern regions. Second, the land use data were prescribed in 2013
due to the limited availability of data, and this may neglect some major land use changes in China over the past decades.
Though the land use data were found by the ML algorithm to contribute little to the overall model, more detailed land use



data are expected to further increase model accuracy. In addition, though concentration-based metrics are easy to calculate
and ensured to be scientifically sound in some experiments (Fuhrer et al., 1997; Mills et al., 2007), they do not consider
the active responses of plant ecophysiology to ambient climatic and environmental changes and thus likely inadequate for
examining yield losses in a future climate and atmospheric environment. Thus, flux-based metrics are recommended in
future studies to better understand the long-term evolution of crop losses over China (Feng et al., 2012; Zhang et al., 2017;
Tai et al., 2021; Pleijel et al., 2022). Despite these limitations, our study represents important progress in evaluating the
long-term, multidecadal health burdens and agricultural losses resulting from $O_3$ pollution in China, which can provide
important references for governments and agencies when making related policies to meet the imperative environment,
health, and food security demands.

**Competing interests**

The contact author has declared that neither they nor their co-authors have any competing interests. At least one of the
(co-)authors is a member of the editorial board of Atmospheric Chemistry and Physics.

**Acknowledgements**

This work was supported by the National Natural Science Foundation of China (NSFC)/Research Grants Council (RGC)
Joint Research Scheme (reference #: N_CUHK440/20, 42061160479) awarded to A. P. K. Tai and Z. Feng.

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
