# Peer review of "Multidecadal ozone trends in China and implications for human"

_EGUsphere, 2023_

## Author Comment (AC1)

**Responses to Reviewers' Comments on "Multidecadal ozone trends in China and implications for human health and crop yields: A hybrid approach combining chemical transport model and machine learning" by Mao et al. (MS No.: acp-2023-1052)**

We would like to thank the reviewers for the thoughtful and insightful comments. The manuscript has been revised accordingly, and our point-by-point responses are provided below. The reviewers' comments are *italicized*, our replies are in black font, and our new/modified text cited below is highlighted in **bold**.

**Response to Referee #1**

*The aims of the presented study are to 1) use improved finer high-resolution hourly ozone data to assess ozone impacts on human health and crop yields over the past four decades in China, and 2) use the findings to offer more comprehensive policy implications for mitigation of ozone-related impacts across China. The research conducted is interesting and beneficial to the agricultural, modeling, and health fields, mainly in China. The study is well described, however, there are some minor issues that hinder the clarity. Minor revision is recommended before acceptance.*

We thank the reviewer for the very helpful comments. The paper has been revised accordingly to address the reviewer's concerns point by point, and all changes are cited and discussed in the responses below.

*Why were the BTH, SCB, YRD, and PRD given more detailed analysis compared to other regions? Which areas correspond more to agricultural production or human health? This should be stated.*

We thank the reviewer for the comments. The BTH, SCB, YRD, and PRD are hotspots of $O_3$ pollution in China mostly due to the high level of industrialization and urbanization. Moreover, these regions are densely populated (Wang et al., 2018) and major agricultural areas in China (Monfreda et al., 2008). These regions may face greater burdens of crop yield and human health losses with high $O_3$ concentrations, and are therefore given more detailed analysis here. We now state these more clearly in the manuscript:

P12 L362: "…The regional characteristics of $O_3$ and its influencing factors will be further discussed in Section 3.4. **The BTH, SCB, YRD, and PRD regions have been identified as hotspots of $O_3$ pollution in China. These regions are characterized by high population density (Wang et al., 2018) and are also major agricultural areas (Monfreda et al., 2008), which may face greater burdens of crop yield and human health losses with high $O_3$ concentrations. Therefore, here we provide more detailed analysis and investigation of these regions.**"

P20 L579: "Despite these limitations, our study represents important progress in evaluating the long-term, multidecadal health burdens and agricultural losses resulting from $O_3$ pollution in China. **Across the four major regions, BTHs experience the highest RYLs for major crops due to elevated $O_3$. On the other hand, the YRD and PRD regions have greater human health losses primarily due to their large population size.**"

*The authors state that the findings can offer more comprehensive policy implications for mitigation of O3-related impacts, but do not mention any policy implication in the Conclusion/discussion section. This should be added to the Conclusion.*

We now discuss the policy implications and possible efforts more fully in the Conclusions and Discussion section.

P20 L582: "…The results can provide important references for governments and agencies when making related national or regional policies to meet the imperative environment, health, and food security demands. **To effectively address O₃ impacts, collaborative efforts can be made in multifaceted aspects: (1) to implement stricter regulations and specific emission control measures for major ozone precursors from industrial, vehicular and agricultural sources that account for region-specific chemical, meteorological and terrestrial conditions; (2) to encourage the adoption of more sustainable and adaptive agricultural practices that minimize O₃ exposure and its damage on crops (e.g., cultivating O₃-resistant crop varieties); (3) to improve short-range O₃ forecast capabilities of regional models, especially with the enhancement of artificial intelligence technology, which may enable better early warning systems to prepare the public and farmers for O₃ episodes; (4) to raise public awareness via promotional campaigns and educational programs to inform individuals, communities, and farmers about the risks associated with O₃. It is important for policymakers to consider these suggestions and act to effectively mitigate the negative O₃ impacts."**

*Specific comments:*

*Lines 181-182: "...datasets at different spatial resolutions were all regridded to a unified resolution of 0.25 x 0.25...". How were they regridded/downscaled/aggregated? Please describe the methods used.*

The method was introduced in revised manuscript.

P5 L177 "…Because the representation of input data for LightGBM should be regular, datasets at different spatial resolutions were all regridded to a unified resolution of 0.25°×0.25° **with the operationally used bilinear interpolation approach (e.g., Accadia et al., 2003)**, consistent with the meteorological fields.

*Line 214: "It has been suggested that, suggesting…" please clarify sentence.*

The sentence was revised as suggested.

*Line 219: Should be referencing Table S2 instead of Table S1. Switch Table S2 and S1 in the supplementary since Table S2 is mentioned first.*

All Table/Figure order and referencing has been checked and revised.

*Line 239: Should be referencing Table S4 instead of S2. All Table/Figure order and referencing should be checked.*

All Table/Figure order and referencing has been checked and revised.

*Figure 2 x-axis label should be "Feature importance". May be better to put label below x-axis instead of above plot.*

The label was added below x-axis as suggested.

*Line 276: Mention RMSE in µg m-3 instead of ppb for consistency with other results presented.*

The unit was changed to µg m⁻³ for consistency.

*Line 393: Should be referencing Table S3 instead of Table S1.*

All Table/Figure order and referencing has been checked and revised.

*Lines 395-396: "Previous studies reported…in the planetary boundary layer (PBL), etc." Cite studies mentioned and remove "etc.".*

The relevant references were added.

*Line 421: Should be referencing Table S3 instead of Table S2.*

All Table/Figure order and referencing has been checked and revised.

*Line 528: Use "RYL" instead of "RLY".*

The same typo was revised through the whole draft.

**Response to Referee #2**

*The manuscript utilized a machine learning algorithm, i.e., LightGBM, to bias-correct surface ozone estimates by the GEOS-Chem model during 1981-2019 in China. The results show that the accuracy of the simulated surface ozone estimates was considerably improved. The authors employed these improved surface ozone estimates to assess the extent to which crop yields and human health were impacted in China. Overall, the topic is of interest to the audience and the manuscript is generally well written and organized. However, before I can only recommend it to be accepted by the EGUsphere journal, the manuscript needs some major revision.*

We thank the reviewer for the very helpful comments. The paper has been revised accordingly to address the reviewer's concerns point by point, and all changes are cited and discussed in the responses below.

*The surface ozone concentration measurements were obtained only for the period 2016-2018, whereas there are longer records. The authors need to clarify why they only adopted observations in such a short period to train and test the LightGBM model.*

We thank the reviewer for the very relevant comment. The surface ozone concentration measurements in China were available since in 2013 with relative scarce sites in the first few years. During the model training process, we found that the training time was approximately linearly related to the number of samples when altering the size of the training dataset, and a timescale of two years appears to strike a good balance between computational burden and utility for an operational system such as air quality forecasting (**Fig. R1**). To optimize computational efficiency without compromising model robustness and accuracy, we utilized observations from the period 2016-2017 as the training data, and observations in 2018 as the independent test data.

[Figure]

**Figure R1.** Testing statistics with increasing length of training data for MDA8-O₃.

We now emphasize these in P6 L203 "…**Our analysis revealed that training the model with one year or more of data results in only marginal reductions in RMSE and enhancements in $R^2$ (Fig. S1); thus a timescale of two years appears to strike a good balance between computational burden and model accuracy. These results align with the findings of Ivatt and Evans (2020), who suggested that much of the variability in the power spectrum of surface O₃ can be captured by timescales of a year or less. Therefore, here we utilized observations from the period 2016-2017 as the training data, which offered a more economical computing cost and improved training time efficiency, and observations in 2018 as the independent test data to evaluate model performance.**"

Fig.R1 is also added as Fig. S1 to the supplementary materials.

*There is a scale mismatch between ground observations and GEOS-Chem estimate for a grid, i.e., there may be multiple ground sites within a 0.25x0.25 grid cell. How did the authors handle this issue?*

We thank the reviewer for the comments. First and foremost, it is worth noting that the observations were only used for the purpose of model evaluation to assess the accuracy and robustness of the model. The handling method is now explained in greater detail:

P6 L197 "…**During evaluation, the model results in the grid cell covering or closest to each site were utilized to compare with observations. This approach of comparing model simulated gridded air pollutant concentrations (either from a CTM or ML model) against site-level observations has been commonly used to evaluate model performance (Ma et al., 2021; Meng et al., 2022; Thongthammachart et al., 2021). Additionally, when comparing the GEOS-Chem-simulated O₃ with observations, the simulated O₃ was first regridded to 0.25°×0.25° using the operationally used bilinear interpolation approach to maintain consistency with the ERA5 dataset.**"

*The GEOS-Chem simulations used the MERRA2 climate dataset, while the LightGBM used ERA5 climate data. The difference between the two climate datasets will be transferred into the LightGBM model training, which potentially*

*impedes the machine learning model to capture the biases between GEOS-Chem estimates and ground observations. The authors need to analyze the uncertainty propagation.*

We thank the reviewer for the comments. To provide long-term GEOS-Chem simulated $O_3$ fields for incorporation into the ML model, we conducted GEOS-Chem simulations at a resolution of 2.0°×2.5° as higher resolutions of GEOS-Chem in nested grids are available but computationally prohibitive for multi-decadal simulations. Therefore the MERRA2 climate dataset used to drive GEOS-Chem also has a resolution of a horizontal resolution of 2.0°×2.5°. We trained the model with MERRA2 dataset; however, the results show the higher-resolution ERA5 dataset performed better in reproducing observed $O_3$ concentrations with smaller RMSE and larger $R^2$ (**Fig. R2**) even though MERRA2 was first regridded to a resolution of 0.25°×0.25° consistent with ERA5 dataset. This analysis demonstrates the level to which higher-resolution meteorological data as opposed to the lower-resolution default MERRA2 data may help enhance the performance of the hybrid model, and the differences can be attributed to the differences in meteorological datasets.

[Figure]

**Figure R2.** Density scatter plots and linear regression statistics of LightGBM bias-corrected $O_3$ predictions vs. observation for 2018: (a) MDA8-$O_3$ using ERA5 meteorology vs. observations; (b) MDA8-$O_3$ using MERRA2 meteorology vs. observations; (c) hourly $O_3$ using ERA5 meteorology vs. observations; and (d) hourly $O_3$ using MERRA2 meteorology vs. observations. The dashed red line indicates the 1:1 line, and the solid blue line indicates the line of best fit using orthogonal regression. The $R^2$ is the coefficient of determination, RMSE is the root-mean-square error, and N is the number of data points. The X and Y axis represents the $O_3$ observations and predictions, respectively.

Because the objective of our study is to reproduce more reliable $O_3$ concentrations using the most comprehensive relevant data as much as possible, the greatest attention is given to the accuracy of the hybrid model rather than the biases of the GEOS-Chem model caused by errors in input data. In summary, with a higher prediction accuracy, the

hybrid approach lends greater credence to using model simulations to extrapolate historical O₃ further back in time, which can furthermore provide us with more accurate estimates of O₃ impacts on crop production and human health.

We now emphasize these in P8 L270 "… **The MERRA2 dataset driving GEOS-Chem was also used to train the model; however, we found that the higher-resolution ERA5 dataset performs better in reproducing observed O₃ concentrations with smaller RMSE and larger $R^2$ (Fig. S3). This analysis demonstrates the level to which a higher-resolution meteorological dataset, despite not being strictly consistent with the input meteorology for the CTM, can help enhance the performance of the hybrid model. In summary,** the result suggests that the CTM-simulated results can be substantially improved by applying ML with multi-source datasets, and the bias-corrected data can improve our understanding of long-term O₃ trends and its further implications on crop and human health over China, as discussed in the following sections.**"**

P19 L515 "… **In the model training process, we found that utilizing a higher-resolution meteorological dataset, albeit one that is not the same as the default CTM input meteorology, has high potential to enhance the performance of the hybrid model in reproducing observed O₃ concentrations."**

Fig. R2 is also added as Fig. S3 to the supplementary materials.

*In the abstract, the manuscript writes that meteorological factors play important roles in modulating the inter-annual variability of surface ozone. However, there is no any evidence (figures or statistics) in the manuscript to support this conclusion.*

We thank the reviewer for pointing this out. That statement is indeed mostly a summary of previous research findings, not a primary focus or finding from this study. To avoid confusion, this statement in the abstract has now been removed.

**References:**
Accadia, C., Mariani, S., Casaioli, M., Lavagnini, A., and Speranza, A.: Sensitivity of Precipitation Forecast Skill Scores to Bilinear Interpolation and a Simple Nearest-Neighbor Average Method on High-Resolution Verification Grids, Weather and Forecasting, 18, 918-932, https://doi.org/10.1175/1520-0434(2003)018<0918:SOPFSS>2.0.CO;2, 2003.

Ivatt, P. D. and Evans, M. J.: Improving the prediction of an atmospheric chemistry transport model using gradient-boosted regression trees, Atmos. Chem. Phys., 20, 8063-8082, https://doi.org/10.5194/acp-20-8063-2020, 2020.

Ma, R., Ban, J., Wang, Q., Zhang, Y., Yang, Y., He, M. Z., Li, S., Shi, W., and Li, T.: Random forest model based fine scale spatiotemporal O3 trends in the Beijing-Tianjin-Hebei region in China, 2010 to 2017, Environ Pollut, 276, 116635, https://doi.org/10.1016/j.envpol.2021.116635, 2021.

Meng, X., Wang, W., Shi, S., Zhu, S., Wang, P., Chen, R., Xiao, Q., Xue, T., Geng, G., Zhang, Q., Kan, H., and Zhang, H.: Evaluating the spatiotemporal ozone characteristics with high-resolution predictions in mainland China, 2013–2019, Environmental Pollution, 299, 118865, https://doi.org/10.1016/j.envpol.2022.118865, 2022.

Monfreda, C., Ramankutty, N., and Foley, J. A.: Farming the planet: 2. Geographic distribution of crop areas, yields, physiological types, and net primary production in the year 2000, Global Biogeochemical Cycles, 22, https://doi.org/10.1029/2007GB002947, 2008.

Thongthammachart, T., Araki, S., Shimadera, H., Eto, S., Matsuo, T., and Kondo, A.: An integrated model combining random forests and WRF/CMAQ model for high accuracy spatiotemporal PM2.5 predictions in the Kansai region of Japan, Atmospheric Environment, 262, https://doi.org/10.1016/j.atmosenv.2021.118620, 2021.

Wang, L., Wang, S., Zhou, Y., Liu, W., Hou, Y., Zhu, J., and Wang, F.: Mapping population density in China between 1990 and 2010 using remote sensing, Remote Sensing of Environment, 210, 269-281, https://doi.org/10.1016/j.rse.2018.03.007, 2018.

---

## Author Response (AR2)

**Responses to Reviewers' Comments on "Multidecadal ozone trends in China and implications for human health and crop yields: A hybrid approach combining chemical transport model and machine learning" by Mao et al. (MS No.: acp-2023-1052)**

We would like to thank the reviewers for the thoughtful and insightful comments. The manuscript has been revised accordingly, and our point-by-point responses are provided below. The reviewers' comments are *italicized*, our replies are in black font, and our new/modified text cited below is highlighted in **bold**.

**Response to Referee #1**

We thank the reviewer for the very helpful comments. The paper has been revised accordingly to address the reviewer's concerns point by point, and all changes are cited and discussed in the responses below.

*Specific comments:*

*The authors addressed all comments satisfactorily but did not correct the plot scales in Figures 4, 6, S2, and S7-S9 so it is difficult to compare the results between the subplots, e.g., (a) vs (c). A single scale should be used for each figure so that readers can make visual comparisons between the subplots. The study is recommended for acceptance after the plot scales are corrected.*

We thank the reviewer for the comments. The scale in each mentioned figure has been revised to a single scale accordingly.

**Response to Referee #2**

*First, the observations are not only used for validation but also for training. Thus, the response "it is worth noting that the observations were only used for the purpose of model evaluation to assess the accuracy and robustness of the model" is problematic. Specifically, the scale mismatch issue is not resolved. The authors should recognize that a site and a grid of 0.25 deg are at different spatial scales, at which observations are not directly comparable.*

We are sorry for the problematic statement. Yes, the observations were used both for training and validation. We intended to explain in our response that during both training and evaluation phases, observational data were not used as predictor variables and were solely utilized as the response variables and for comparison with model results. We also fully acknowledge that spatial scale mismatch is a common problem for model-observation comparison, and now discuss so more fully. We emphasize that the whole purpose of using machine learning (ML) here and in other similar studies is to minimize the biases of model output, whereby the biases can arise from incomplete model physics, input and parameter errors, numerical errors, coding errors, as well as representation errors (i.e., mismatch in spatial scales between model output and observations, as the reviewer pointed out), so that the output of the ML-enhanced hybrid model can be the closest to the observations for more accurate impact evaluation. That is, the biases arising from the spatial scale issue has indeed been considered and inherently addressed in our bias reduction approach. See below for our modified text:

P5 L176: "**The primary purpose of utilizing ML here was to minimize the biases of model output as compared with observations, whereby the biases could arise from incomplete model physics, input and parameter errors, numerical errors, coding errors, as well as representation errors (i.e., mismatch in spatial scales between model**

**grid cells and site observations), so that the output of the hybrid model could have the closest values to the observations and enable more accurate impact evaluation.** In this study, we used the LightGBM ML algorithm to integrate GEOS-Chem-simulated $O_3$ at a lower resolution with higher-resolution multi-source data to produce higher-resolution hourly $O_3$ and MDA8-$O_3$ fields. … (P5 L187) **The training and evaluation processes are both performed at the site level in accordance with the observations, whereby the predictor variables and model responses were first sampled at the same locations using the bilinear interpolation approach (Accadia et al., 2003). This approach of handling spatial scale mismatch between model grid cells and site observations has been commonly used in previous studies (e.g., Li et al., 2021). When predicting the gridded $O_3$ concentrations with the trained model, predictor variables at different spatial resolutions were all regridded to the same resolution of 0.25°×0.25° consistent with the ERA5 meteorological fields.** …"

*Second, as shown in Fig. R2, it is doubtful whether it is necessary to use ERA5 to downscale simulated O3 concentrations. The results based on ERA5 showed no significant improvements over those based on MERRA2.*

We thank the reviewer for the comments. To respond to the reviewer's concern, we have extensively conducted additional analysis using two separate meteorological datasets (ERA5 & MERRA2) for model training to investigate into whether using a higher-resolution meteorological fields truly brings benefits to the outcomes in terms of accuracy. We have shown the comparison of the performance between the MERRA2 and ERA5 datasets in the supplementary materials. Ultimately, we selected the results using the ERA5 dataset for further analysis due to its moderately higher accuracy in terms of the considered statistical metrics (e.g., as shown in **Fig. S3**, for MDA8 $O_3$, $R^2$ increases from 0.69 to 0.72, and RMSE decreases from 25.21 to 23.76 $\mu g\ m^{-3}$) as well as the inclusion of more refined spatial details within the original GEOS-Chem grid cells, because the primary objective of the bias correction process to achieve the highest-possible level of accuracy in ozone concentration estimation for further impact analysis. We now elaborate these points further in the revised manuscript:

P5 L191: "… **When predicting the gridded $O_3$ concentrations with the trained model, predictor variables at different spatial resolutions were all regridded to the same resolution of 0.25°×0.25° consistent with the ERA5 meteorological fields. By taking the advantage of these higher-resolution datasets, the hybrid approach can not only correct the biases of the GEOS-Chem-simulated $O_3$, but also refine them into a finer resolution. To evaluate if the hybrid approach truly benefits from using a higher-resolution meteorological fields, we also repeated the whole training exercise with the input meteorology of GEOS-Chem (MERRA2 at 2.0°×2.5°) instead of ERA5.**"

P8 L272: "… **To test if using the higher-resolution meteorological data offers better prediction accuracy compared with the original input meteorology of GEOS-Chem, the MERRA2 dataset driving GEOS-Chem was also used to train the model. We found that the higher-resolution ERA5 dataset performed better in reproducing observed $O_3$ concentrations with moderately smaller RMSE and larger $R^2$ (Fig. S3), demonstrating the level to which a higher-resolution meteorological dataset, despite not being strictly consistent with the input meteorology for the CTM, can help enhance the performance of the hybrid approach and help resolve finer spatial details within the original CTM grid cells.** In summary, …"

Accadia, C., Mariani, S., Casaioli, M., Lavagnini, A., and Speranza, A.: Sensitivity of Precipitation Forecast Skill Scores to Bilinear Interpolation and a Simple Nearest-Neighbor Average Method on High-Resolution Verification Grids,

Weather and Forecasting, 18, 918-932, https://doi.org/10.1175/1520-0434(2003)018<0918:SOPFSS>2.0.CO;2, 2003.

Li, K., Jacob, D. J., Liao, H., Qiu, Y., Shen, L., Zhai, S., Bates, K. H., Sulprizio, M. P., Song, S., Lu, X., Zhang, Q., Zheng, B., Zhang, Y., Zhang, J., Lee, H. C., and Kuk, S. K.: Ozone pollution in the North China Plain spreading into the late-winter haze season, Proceedings of the National Academy of Sciences, 118, e2015797118, https://doi.org/10.1073/pnas.2015797118, 2021.